# 8-Oxo-7,8-Dihydro-2′-Deoxyguanosine (8-oxodG) and 8-Hydroxy-2′-Deoxyguanosine (8-OHdG) as a Potential Biomarker for Gestational Diabetes Mellitus (GDM) Development

**DOI:** 10.3390/molecules25010202

**Published:** 2020-01-03

**Authors:** Sandra K. Urbaniak, Karolina Boguszewska, Michał Szewczuk, Julia Kaźmierczak-Barańska, Bolesław T. Karwowski

**Affiliations:** DNA Damage Laboratory of Food Science Department, Faculty of Pharmacy, Medical University of Lodz, ul. Muszynskiego 1, 90-151 Lodz, Poland; sandra.urbaniak@stud.umed.lodz.pl (S.K.U.); karolina.boguszewska@stud.umed.lodz.pl (K.B.); michal.szewczuk@stud.umed.lodz.pl (M.S.); julia.kazmierczak-baranska@umed.lodz.pl (J.K.-B.)

**Keywords:** gestational diabetes mellitus, 8-hydroxy-2′-deoxyguanosine, 8-oxo-7,8-dihydro-2′-deoxyguanosine, oxidative stress, DNA damage

## Abstract

The growing clinical and epidemiological significance of gestational diabetes mellitus results from its constantly increasing worldwide prevalence, obesity, and overall unhealthy lifestyle among women of childbearing age. Oxidative stress seems to be the most important predictor of gestational diabetes mellitus development. Disturbances in the cell caused by oxidative stress lead to different changes in biomolecules, including DNA. The nucleobase which is most susceptible to oxidative stress is guanine. Its damage results in two main modifications: 8-hydroxy-2′-deoxyguanosineor 8-oxo-7,8-dihydro-2′-deoxyguanosine. Their significant level can indicate pathological processes during pregnancy, like gestational diabetes mellitus and probably, type 2 diabetes mellitus after pregnancy. This review provides an overview of current knowledge on the use of 8-hydroxy-2′-deoxyguanosineand/or 8-oxo-7,8-dihydro-2′-deoxyguanosine as a biomarker in gestational diabetes mellitus and allows us to understand the mechanism of 8-hydroxy-2′-deoxyguanosineand/or 8-oxo-7,8-dihydro-2′-deoxyguanosine generation during this disease.

## 1. Introduction

One of the major complications of pregnancy is gestational diabetes mellitus (GDM) [1]. According to global guidelines, GDM may be defined as being any hyperglycemic state that occurs during the second half of gestation among previously healthy pregnant women [2,3,4]. GDM is usually diagnosed based on positive one-step 75-g oral glucose tolerance test (OGTT), when there are no additional factors that might indicate occurrence of prior type 1 or 2 diabetes mellitus (T1DM and T2DM, respectively) [3,4].

The worldwide prevalence of GDM is estimated to range from 3% to 5% and this number is constantly growing, mainly due to increasing obesity and physical inactivity [5]. Diabetes seems to be more common in non-Caucasian populations such as: African American, Hispanic American, Native American, Pacific Islander, and South or East Asian populations [6]. Other factors include an increased body mass index (BMI), family history of diabetes, maternal age over 25, and previous GDM [7,8].

Even though GDM normally declines post-partum, a greater number of patients who are undiagnosed or diagnosed later develop T2DM following pregnancy, cardiovascular disease, or metabolic syndrome [9,10]. Hyperglycemic state also positively correlates with an increased fetal infection rate and mortality. Offspring, like their mother, in adulthood, have a greater predisposition to develop T2DM, metabolic syndrome, and obesity [11].

Although many studies on the pathophysiology of GDM have been conducted, the precise mechanism of its development remains unclear. Hyperglycemia occurring in GDM women is probably the result of β-cells progressive dysfunction [12]. Growing evidence indicates inflammation [13,14], disruption of the insulin signaling pathway [15], plasma adipokine levels alteration [16], and endoplasmic reticulum stress [14] as some of the reasons for the above process. However, oxidative stress (OS) is considered to be the primary cause of GDM development and its components should be strictly controlled during pregnancy and treatment [17].

Purines are organic compounds which can undergo oxidation reaction and form different products [18]. Among the most important biological markers of OS the following ones belonging to purines may be distinguished: 8-hydroxy-2′-deoxyguanosine (8-OHdG) or its oxidized form—8-oxo-7,8-dihydro-2′-deoxyguanosine (8-oxodG). Although all living cells develop a broad range of DNA repair mechanisms, their enzymatic repair system does not always lead to a complete removal of all DNA modifications. Therefore, misrepaired DNA constitutes a major problem for cells, mainly because of genetic information changes as well as mutagenesis and cell apoptosis connected with them [19]. Growing evidence proves that accumulation of abundant lesions, mainly 8-OHdG or 8-oxodG, is an important factor indicating development of GDM, and in the aftermath, T2DM in mothers and their offspring [20,21].

This review summarizes the current knowledge about the impact of oxidative guanine (G) base damages, mainly 8-OHdG or its oxidized form 8-oxodG on GDM development. Additionally, biomarkers of GDM and several mechanisms connected with OS induction and GDM development are described.

## 2. The Influence of ROS Overproduction on GDM Development

Reactive oxygen species (ROS) are highly reactive derivatives of oxygen molecules arising as a result of incomplete oxygen reduction. Their high reactiveness is determined by the existence of at least one unpaired electron on the valence shell [22].

About 5% of the total inhaled oxygen is converted into ROS. Those most important ones are, among others: superoxide radical (O_2_^•−^), hydrogen peroxide (H_2_O_2_), hydroxyl radical (^•^OH), and singlet oxygen (^1^O_2_) [23] ROS are produced mainly in response to radiation (alfa, beta, gamma, X-radiation (X-ray)) or Ultraviolet-Visible (UV-Vis), inflammation (infections), chronic diseases (alcoholism, injuries, cancer), chemical compounds (pesticides, benzopyrene, nitrous oxide), metabolic processes (peroxidation of fatty acids), and metabolic disorders (diabetes mellitus). Moreover, ROS are derived from normal physiological processes conducted in various cellular compartments [24]. In physiological conditions, these processes maintain ROS at the right level and provide proper reduction and oxidation reactions in the respiratory chain, transport of oxygen by hemoglobin, regeneration of energy sources, phagocytosis process, gene expression regulation, and activation of cytochrome P450 [25].

However, excessive production of ROS and insufficient work of antioxidant defense mechanism can lead to a condition known as oxidative stress [25,26]. Oxidative stress is defined as an imbalance towards prooxidative changes which leads to damage and dysfunction of cellular structures, cells and whole organisms. This imbalance is associated with overproduction of free radicals in mitochondria, with formation of advanced glycation end products (AGEs) or as a result of activation of protein kinase C (PKC), polyol or hexoamine pathway in response to elevated glucose levels [27]. Increased ROS production is often accompanied by reduced activity of antioxidant systems, which involves deepening of the OS (Figure 1) [25,27].

In an uncomplicated pregnancy, the level of oxidative stress and lipid peroxidation is higher than normal, mainly because of the mitochondria-rich placenta. Nonetheless, most of body cells properly stimulate the production and activation of antioxidants and, ultimately, maintain the ROS concentration at the safe level. In the case of a pregnancy-related diabetes complication, the observed oxidative stress is the result of hyperglycemia [17]. The impossibility of metabolizing an excess of glucose leads to the condition in which all of above-mentioned pathways are activated [8].

Protein glycosylation is a nonenzymatic process of modification of both intracellular and extracellular proteins [28]. The resulting AGEs modify proteins [29] and, by interacting with specific receptors for advanced glycation end products (RAGEs), activate an inflammatory response and induce oxidative stress [30] as well as insulin resistance [29].

The high blood glucose concentration activates the PKC pathway and consequently leads to inhibition of endothelial nitric oxide synthase (eNOS) [31], activation of NADPH Oxidase 5 (NOX5) [32], stimulation of superoxide anion formation [33], and mediation of fatty acid-induced β-cell apoptosis [34].

Another activated pathway—the polyol pathway, leads to the reduction of glucose in fructose 3-phosphate (fructose-3-P) caused by sorbitol dehydrogenase. As a consequence, this leads to sorbitol-induced osmotic stress, reduction of Na^+^/K^+^ ATPase activity, a decrease of NADPH/NADP ratio [35,36], and finally reduced antioxidant capacity of the glutathione system [36].

Activation of the hexosamine pathway, the third of the mentioned pathways, causes accumulation of nucleotide sugar—Uridine diphosphate *N*-acetylglucosamine (UDP-GLC-NAc) and its oxygen derivatives. UDP-GLC-NAc leads to pathological expression of transforming growth factor-beta (TGFbeta1) genes [37].

The effect of ROS on cellular processes mainly depends on the strength and duration of exposure. The destructive action of radicals can include almost all of the biomolecules which occur in the body—protein modifications, lipid peroxidation, and DNA mutations [23,25]. Ultimately, cells abandon the cell cycle and enter the G0 phase or, in the case of permanent exposure and/or a high concentration of ROS, activate the process of programmed cell death [38]. 

## 3. Types of the Antioxidant Defense Mechanisms

Organisms have created various integrated antioxidant defense mechanisms which neutralize or reduce the negative effects of ROS and lead to the “oxidative balance”. The proper concentration of antioxidants is necessary for maintaining basic cell functions (proliferation, differentiation, energy production) and a whole organ functioning at the right level and its activity can change depending on intracellular concentration of ROS. The antioxidative defense system includes enzymatic (superoxide dismutase (SOD), catalase (CAT), thioredoxin (TRX), peroxiredoxin (PRX), heme oxygenase-1, glutathione peroxidase (GPx), glutathione reductase (GRd), and glutathione S-transferase (GST)) and non-enzymatic (low molecular weight) (all-trans retinol 2 (Vitamin A), ascorbic acid (Vitamin C), α-Tocopherol (Vitamin E), β-Carotene, uric acid, glutathione (GSH), and tripeptide (l-γ-glutamyl-l-cysteinyl-l-glycine)) antioxidants [39].

## 4. Biological Markers of Oxidative Stress in GDM Women

The degree of oxidative stress among women with GDM can be measured using different biomarkers [20,40] which are classified according to the reactions that change them [41]. Two classes of biological markers may be distinguished: ROS-modified molecules and antioxidant molecules altered by increased redox stress [41].

### 4.1. Influence of Oxidative Stress on Lipids

The main indicators of increased oxidative stress in diabetes, as well as in patients with GDM, are lipid peroxidation products that increase fluidity and permeability of the cell membrane [42,43]. Studies show increased levels of malonyl dialdehyde (MDA) in the placenta tissue, cord plasma, and maternal plasma among patients with GDM [44]. MDA reacts with thiobarbituric acid (TBA) which leads to the formation of thiobarbituric acid reactive substances (TBARS) [27]. Higher level of TBARS is observed in serum from diabetic mothers and their macrosomic offspring [45] as well as in cord blood of newborns [46]. Among lipid peroxidation products, an important indicator is also an increased concentration of lipid hydroperoxide (LOOH) in serum [47] and 8-iso-PGF2α, belonging to F2-isoprostanes [40,48]. TBARS and 8-iso-PGF2α are used as an indirect biomarkers of OS [39].

### 4.2. Influence of Oxidative Stress on Proteins

Besides lipid peroxidation, protein oxidation is also presented under GDM conditions. OS can trigger formation of protein cross-linking, fragmentation of the peptide as well as formation of modified, denaturated, and non-functioning proteins as well as intensified proteolysis reactions [39]. Moreover, a hyperglycemic state can also lead to protein glycation [49]. Among patients with GDM, higher levels of advanced oxidation protein products (AOPP) [50,51], protein hydroperoxides (POOHs), protein carbonyls (PCO) [51], *C*-reactive protein (CRP) [40], and glycated hemoglobin (HbA1c) can be presented [51]. Moreover, ROS can also decrease the level of Paraoxonase 1 (PON1) [51] and, depending on the conducted clinical studies, cause an increased [51] or decreased [52] level of 3-nitrotyrosine (3-NT). Increased formation and accumulation of protein oxidation products is considered as an important mediators of adipocyte disorders. They intensify inflammatory response and play a significant role in chronic development of diabetic complications [28].

### 4.3. Influence of Oxidative Stress on Enzymatic and Non-Enzymatic Antioxidants

Considering the damage caused by oxidative stress in GDM, the most frequently studied enzymes have been superoxide dismutase (SOD), catalase (CAT), and glutathione peroxidase (GPX). Current data indicate that the concentration of each of these enzymes is decreased [53]. 

In case of vitamins, the results are less clear. Concentration of individual vitamins are either reduced (vitamin C and vitamin E) [54], increased (vitamin C [45] and vitamin E [55]), or remains at the same level (vitamin A) [45]. Probably, these discrepancies can be a result of different diagnostic criteria and/or too small examined samples [27].

### 4.4. Influence of Oxidative Stress on DNA

Oxidative damages of nucleic acids are the most dangerous modifications observed among biomolecules. They can be distinguished into: oxidation of bases and/or sugar fragments, single/double strand breaks (SSB and DSB, respectively), basic/apurinic/apirimidinic (AP) sites, purine/pyrimidine/sugar-related modifications, structural mutations of nucleic bases, and DNA-protein cross-linking, deletions and/or translocations of chromosome fragments. The accumulation of such damages can lead to changes of genetic information, and consequently to mutagenesis and cells apoptosis [56].

The hydroxyl radical ^•^OH is responsible for the majority of DNA damages. It can attack the C5 carbon atom or carbon atom from the methyl group (CH3) presented in the pyrimidines, which is the C8 atom in the case of purines or the amino group of adenines (A) [57]. 

Of all the nucleic bases, G is the most susceptible to oxidative stress caused by ROS. The best-known damage is 8-Oxoguanine (8-oxoG). The latest data indicate that up to 10^5^ 8-oxoG modifications are created daily in DNA per cell. The widespread occurrence of nucleobases damage is a convenient marker of DNA oxidative damage, repair, and cellular oxidative stress in general [56]. GC rich sequences presented at a transcription factor binding sites appear to be the most susceptible to damage. The appearance of 8-oxoG modification at these sites alter the expression of related with them genes [58].

Apart from 8-oxoG, other DNA lesions can be also observed, for instance 5-hydroxymethyl uracil (5-OHMeUra) or 8,5′-cyclo-2′-deoxyadenosine (cyclo-dA) [39]. The first arises as a result of 5-methylcytosine (5-MeCyt) oxidation and can introduce incorrect DNA methylation patterns, which as a consequence lead to gene silencing, a disturbance in chromatin organization, and an incorrect DNA repair mechanism [39,59]. Cyclo-dA appearing in the TATA sequence limits the interaction of proteins with this sequence, which also leads to inhibition of gene transcription [60]. 

## 5. Generation of the Most Abundant Lesions Observed in GDM: 8-OHdG and 8-oxodG 

Modifications of 8-OHdG and 8-oxodG arise as a result of the interaction between ^•^OH or ^1^O_2_ and G of the DNA strand. Free radicals attack G or free 2′-deoxyguanosine, which consequently generates radical adducts. Electron abstraction forms 8-OHdG, which through a reaction known as keto-enol tautomerism, is transformed into the major oxidized product 8-oxodG (Figure 2) [61].

Simone et al. [62], have provided the first likely model of DNA damage formation in diabetes mellitus. According to their results, hyperglycemia can induce redox-dependent activation of serine/threonine-specific protein kinase—Akt which enhances phosphorylation of tuberin protein and as a consequence also enhances downregulation of human 8-oxoguanine-DNA glycosylase 1 (hOGG1), enzyme engaged in the DNA base excision repair pathway (BER).

## 6. Repair Mechanism of 8-OHdG and 8-oxodG in Diabetes Mellitus

Both 8-OHdG and 8-oxodG are recognized by cells as heavy lesions which need to be removed quickly. The foregoing modifications can induce transversion mutations: GC→AT, which can lead to cancer [19] or trigger high levels of oxidative stress [20]. The BER repair system, which was invented by Tomas Lindahl [63], and two executive enzymes: hOGG1 [56] and human MutT homologue (hMTH1), are responsible for removing these alterations [64].

HOGG1 is a bifunctional DNA glycosylase with lyase activity which is responsible for recognizing and the excising of 8-oxodG from the oxidatively-damaged DNA. The general repair mechanism initiated by hOGG1 includes lesion identification, base excision, phosphodiester bond 3′ cleaving, and site-specific changes in the double-helix structure of DNA [56].

hMTH1, also known as 2-Hydroxy-dATP diphosphatase or nudix hydrolase 1 (NUDT1), is an enzyme primarily localized in the cell cytoplasm that performs two functions during the DNA repair process. In the first stage, hMTH1 is responsible for oxidized purines hydrolyzation. It hydrolyzes 8-oxo-2′-deoxyguanosine-5′-triphosphate (8-oxo-dGTP) or 8-oxo-2′-deoxyguanosine-5′-diphosphate (8-oxo-dGDP) to 8-oxo-2′-deoxyguanosine-5′-monophosphate (8-oxo-dGMP), which are not substrates in DNA polymerization [65]. 

## 7. Concentration of 8-OHdG/8-oxodG and Risk of GDM Development

There is no clear evidence of a relationship between DNA damage and risk of glucose intolerance in pregnancy or GDM development. Nonetheless, has been is observed that among patients with GDM or mild gestational hyperglycemia (MGH), a higher level of oxidative DNA damage is presented (Table 1). 

There is growing evidence which indicates that a high level of oxidized purines reflects DNA damage triggered by hyperglycemia. Among patients with MGH (characterized by hyperglycemia, high insulin resistance indices (HOMA-IR), obesity and hypertension) elevated levels of oxidized pyrimidines are recorded [66]. According to Collins et al. [67], the results suggest that various enzymes react differently with oxidized nucleobases (purines or pyrimidines) in various conditions. Formamidopyrimidine DNA glycosylase (FPG) sensitive sites, are increased in women with higher glycemic levels, but not in women with MGH. Patients with MGH, obesity, hypertension, and hyperglycemia presented higher levels of endonuclease III sensitive sites. An increased level of endonuclease III sensitive (Endo III)-sites reflect an overall oxidative damage of DNA, probably induced by hypertension and other side effects of diabetes, whereas FPG-sensitive sites indicate specific damage—8-oxo-guanine triggered by hyperglycemia [66,67]. The results obtained by Gelaleti et al. [66] show that the oxidative DNA damage level (oxidized purines) in lymphocytes of women with GDM was about 70%, whereas the equivalent among the control group was about 65%, with statistical significance *p* < 0.05. Furthermore, the urine concentration of 8-OHdG in GDM women and MGH was marginally higher than in the control group, however, with no statistical significance *p* > 0.05. 

In a pilot study Qiu et al. [20] revealed that an increased level of urine 8-OHdG concentrations correlated with a higher risk of GDM development. Moreover, the level of 8-OHdG in urine among patients with a GDM total of >8.01 ng/mg of creatinine and risk of developing GDM was 3.79-fold higher than for women whose concentration of 8-OHdG in urine was <4.23 ng/mg creatinine. 

Apart from that, the intracellular oxidative status in women during pregnancy also correlated with the fate of a fetus. It seems that a fetal organism is more immune to DNA damage and cell apoptosis induced by oxidative stress than a mother. However, it is possible that there is a limit beyond which the fate of the fetus is precipitously changed [71]. 

Animal studies confirm that general DNA damage in leukocytes from diabetic female rats (mothers) and their fetuses was higher in comparison to control group [68,69,70]. Moreover, Lima et al. [68] reported that (i) among rats with severe diabetes during pregnancy hyperglycemia induced oxidative DNA damage which was identified by FPG or Endo III enzymes, (ii) offspring from rats with severe diabetes presented a higher level of 8OHdG and/or 8-oxodG, (iii) rats with mild diabetes and their newborns presented a higher level of 8OHdG and/or 8-oxodG, which can confirm the fact that FPG sensitive sites do not depend on oxidative stress but instead on induced hyperglycemia.

These results may suggest that oxidative DNA damage can reflect concentration of blood glucose and intensity of diabetes mellitus. Therefore, previous observations allow us to conclude that GDM is a very early stage of diabetes which may evolve into T2DM in the future [4]. 

## 8. Maintenance of Genomic Integrity during GDM

Currently, it is known that maternal hyperglycemia during pregnancy can lead to the formation of oxidative DNA modification, mainly in maternal genetic material [71]. In order to protect and maintain DNA stability, organisms develop a cellular mechanism known as DNA damage response (DDR). Cells with incorrect DDR have a higher risk of mutagenesis, cell death, and diseases development, like cancer, neurodegenerative disorders, cardiovascular disease, and finally metabolic syndrome [72]. Studies confirm the importance of the DNA repair mechanism in pregnancy and fetus development. Among patients whose fetus had DNA repair diseases such as trichothiodystrophy (TTD) or xeroderma pigmentosum D (XPD), an increased risk to deliver preterm, or develop preeclampsia as well as HELLP (hemolysis, elevated liver enzymes, and low platelet count) syndrome was observed. Moreover, fetus movement was decreased [73]. Studies conducted on nonpregnant patients with T1DM and T2DM (both children and adults) showed that patients with T2DM have a higher level of oxidative DNA lesions and decreased efficiency of DDR mechanism compared with other diabetic subjects. Moreover, children with T1DM have a reduced level of DNA and increased efficiency of DDR mechanism [74]. These findings indicate differences in the efficiency of DNA repair mechanism between T1DM and T2DM subjects, which can be associated with long-term consequences. It suggests that the damaging effect of hyperglycemia can be reversed by using DNA repair mechanism. Nonetheless, disturbance in this mechanism can lead to the abnormal maternal-fetal interface [71].

The clinical intervention strategy used to maintain optimal glucose level is based on lifestyle changes including the diet therapy (eating regular meals with low glycemic index (LGI), avoid drinks and food with sugar) and increased physical activity [6,75]. The important thing in GDM treatment is also self-monitoring glucose (fasting glucose and after 1 and 2 h as well as after every meal) [75]. There are different recommendations for nonpregnant diabetics and patients with GDM. According to the American Diabetes Association (ADA), the maternal fasting glucose concentration for the latter should be maintained at the following thresholds values: <95 mg/dL, <140 mg/dL after 1-h, or <120 mg/dL after 2-h [76]. Apart from that, some research suggess that high-dose supplementation of antioxidants in pregnant diabetic rats can regulate the development of their newborns but this treatment is not recommended for nondiabetic pregnancy [77]. Collected evidence indicates that vitamin D can increase insulin secretion and reduce systemic inflammation [78]. Moreover, supplementation of vitamin C during pregnancy can induce preterm delivery but not stillbirth or perinatal death [79]. When lifestyle changes will not provide the expected results, then insulin administration is required. However, this option is only recommended for pregnant women who strggle to maintain the right glucose concentration level [80]. Results obtained by Brown et al. suggest that insulin administration can cause a higher risk of hypertensive disorders development and probably preterm delivery [81]. There is still no clear evidence for the effect of insulin and antioxidant treatment on plasma 8-OHG and tissue 8-OHdG [82]. Despite this fact, controlling of glucose concentration seems to be the most important factor in DNA protection not only because of diabetes and associated with its deleterious effects, but also because of the possibility of developing cancer [83].

## 9. 8-OHdG/8-oxodG as a Potential Biomarker for GDM—Summary

Gestational diabetes mellitus is associated with numerous health complications for a mother and her fetus. Although the mechanism of GDM development remains unclear, oxidative stress seems to be an important factor contributing to development of GDM. Current evidence suggests that even a mild form of hyperglycemia can induce oxidative damage of maternal DNA, which may be observed as an elevated level of 8-OHdG/8-oxodG. Moreover, a higher level of 8-OHdG/8-oxodG alteration in early gestation can be an important factor stimulating gestational diabetes mellitus development, whereas a high level of 8-OHdG or 8-oxodG, which remains after pregnancy, can induce type 2 diabetes mellitus development later. Based on novel data, analysis of urine 8-OHdG concentration can be an important biomarker of GDM mainly due to utilizing statistically significant results, providing an easy way to obtain a test sample, and the fact that urinary excretion of 8-OHdG is not associated with human diet. According to a study performed by Qiu et al. [20], 8-OHdG concentrations of ≥8.01 ng/mg creatinine can be probably a significant indicator of oxidative stress and consequently, of higher risk of GDM development. However, this possible cut-off value should be more closely examined before later clinical application, and should be integrated with insulin sensitivity indices. 

Therefore, future studies should focus on deepening knowledge on 8-hydroxy-2′-deoxyguanosine/8-oxo-7,8-dihydro-2′-deoxyguanosine as a potential biomarker for patients with mild gestational hyperglycemia, before GDM development. To achieve a better understanding of the etiology of this disease and its pathogenesis, it is necessary to select high-risk patients and examine the correlation between concentration of 8OHdG/8-oxodG alteration and progression from gestational diabetes mellitus to type 2 diabetes mellitus.

## Figures and Tables

**Figure 1 molecules-25-00202-f001:**
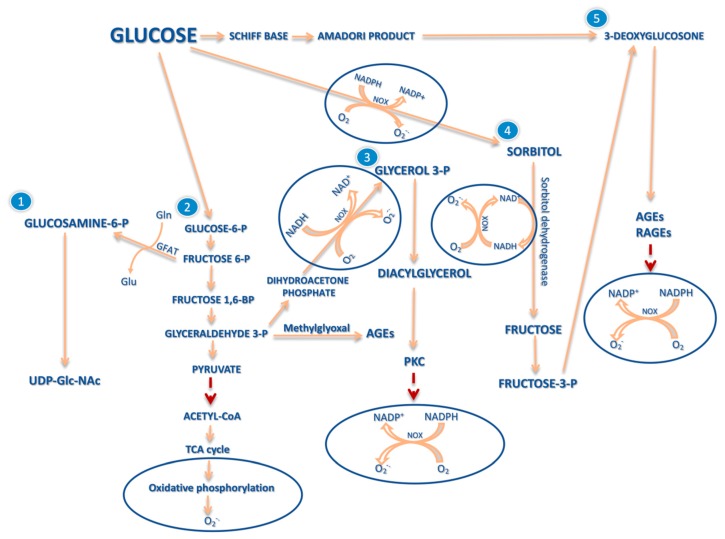
Activation of alternative pathways for glucose metabolism [27]. In a hyperglycemic state, alternative glucose pathways are activated. A higher than normal level of glucose can induce polyol (4), hexoamine (1) and PKC pathways (3), increase basic glycolysis (2), as well as enhance AGEs production (5). Acetyl coenzyme A (**Acetyl-Coa**), advanced glycation end products (**AGEs**), glutamine (**Gln**), glutamic acid (**Glu**), glutamine:fructose-6-phosphate amidotransferase (**GFAT**), NAD(P)H oxidase (**NOX**), nicotinoamide adenine dinucleotide (**NAD^+^/NADH**), protein kinase C (**PKC**), receptors for advanced glycation end products (**RAGEs**), superoxide radical (**O2^•−^),** tricarboxylic acid cycle (**TCA cycle**), uridine diphosphate N-acetylglucosamine (**UDP-Glc-NAc**).

**Figure 2 molecules-25-00202-f002:**
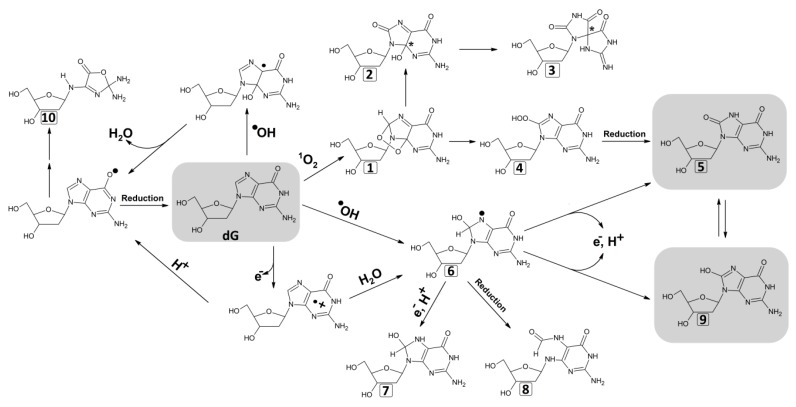
Generation of 8-oxo-7,8-dihydro-2′-deoxyguanosine and 8-Hydroxy-2′-deoxyguanosine [61]. 2′-deoxyguanosine (**dG**), 4,8-endoperoxide-2′-deoxyguanosine (**1**), 4*R*/4*S* 4-Hydroxy-8-oxo-4,8-dihydro-2′-deoxyguanosine (**2**), 4*R*/4*S* spiroiminodihydantoin nucleoside (**3**), 8-Hydroperoxy-2′-deoxyguanosie (**4**), 8-oxo-7,8-dihydro-2′-deoxyguanosine (**5**), 8-Hydroxy-7,8-dihydro-2′-deoxyguanosyl radical (**6**), 7-Hydro-8-Hydroxy-2′-deoxyguanosine (**7**), 2,6-diamino-4-Hydroxy-5-formamidopyrimidine (FapyGua) (**8**), 8-Hydroxy-2′-deoxyguanosine (**9**), oxazolone (**10**).

**Table 1 molecules-25-00202-t001:** DNA damage in hyperglycemic state during pregnancy.

Type of Diabetes	Study Type	Sample	Result	Reference
MGH diabetic women with obesity and hypertension	Clinical	Maternal lymphocytes	↑ Overall oxidative DNA damage	[66]
GDM	Clinical	Maternal lymphocytes	↑ 8-oxoG	
MGH	Clinical	Maternal urine	↑ 8-OHdG	
GDM	Clinical	Maternal urine	↑ 8-OHdG	
GDM	Clinical	Maternal lymphocytes	↑ 8-oxoG	[67]
GDM	Clinical	Maternal urine	↑ 8-OHdG	[20]
MGH, induced by streptozotocin	Experimental	Maternal and fetal leukocytes	↑ 8-OHdG ↑ 8-oxo-dG	[68]
Severe, induced by streptozotocin	Experimental	Maternal and fetal leukocytes	↑ Overall oxidative DNA damage	
Severe, induced by streptozotocin	Experimental	Maternal leukocytes	↑ Overall oxidative DNA damage	[69]
Severe, induced by streptozotocin	Experimental	Fetal leukocytes	↑ Overall oxidative DNA damage	[70]

GDM: Gestational Diabetes Mellitus; MGH: Mild Gestational Hyperglycemia.

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
