# Peer review of "8-Oxo-7,8-Dihydro-2′-Deoxyguanosine (8-oxodG) and 8-Hydroxy-2′-Deoxyguanosine (8-OHdG) as a Potential Biomarker for Gestational Diabetes Mellitus (GDM) Development"

_molecules, 2020, doi:10.3390/molecules25010202_

Round 1
Reviewer 1 Report
The paper review by Urbaniak et al. entitled “8-oxo-7,8-dihydro-2-deoxyguanosine (8-oxodG) and 8-hydroxydeoxyguanosine (8-OHdG) as a potential biomarker for Gestational Diabetes Mellitus (GDM) development” summarizes the link between oxidative deoxyguanosine bases and Gestational Diabetes Mellitus (GDM). The review begins with an introduction about diabetes and oxidative stress. The review then focusses on ROS overproduction on GDM development and highlights oxidative stress on DNA damages. The review ends with a summary of quantitative data of two oxidative DNA damage modifications 8-OHdG and 8-oxodG in humans.
This review provides a comprehensive and nice overview over the link between oxidative DNA damage and diabetes mellitus. I recommend publication of this review in Molecules after major changes.
The structures in Figure 2 are of a too low quality with incorrect bond lengths and format. These must be optimized prior to publication. The review ends abruptly. I strongly suggest to include additional information in part 8: genomic integrity during GDM. The summary is more or less not existing and must be enhanced, I also recommend to use the abbreviations for the DNA bases. This review omits many outstanding researchers in oxidative damage that must be cited before this review is published. There are several additional publications required to be cited for elucidation and investigations of oxidative DNA damages.
Author Response
Dear Reviewer,
Thank you for the review of our paper. We have answered each of the points below.
The paper review by Urbaniak et al. entitled “8-oxo-7,8-dihydro-2'-deoxyguanosine (8-oxodG) and 8-hydroxy-2'-deoxyguanosine (8-OHdG) as a potential biomarker for Gestational Diabetes Mellitus (GDM) development” summarizes the link between oxidative deoxyguanosine bases and Gestational Diabetes Mellitus (GDM). The review begins with an introduction about diabetes and oxidative stress. The review then focusses on ROS overproduction on GDM development and highlights oxidative stress on DNA damages. The review ends with a summary of quantitative data of two oxidative DNA damage modifications 8-OHdG and 8-oxodG in humans.
This review provides a comprehensive and nice overview over the link between oxidative DNA damage and diabetes mellitus. I recommend publication of this review in Molecules after major changes.
The structures in Figure 2 are of a too low quality with incorrect bond lengths and format. These must be optimized prior to publication.
Response: We have made corrections in Figure 2. The quality, bond length and format were optimized. We do hope that now the Figure 2 will be clearer for readers and better illustrate scheme of 8-oxo-7,8-dihydro-2-deoxyguanosine and 8-hydroxydeoxyguanosine generation.
The review ends abruptly. I strongly suggest to include additional information in part 8: genomic integrity during GDM.
Response: This was extremely helpful feedback and we have tried to meet your expectations and improved our work by adding information about: (i) importance of DNA repair mechanism in pregnancy and fetus development, (ii) differences in the DNA damage response mechanism among patients with type 1 and 2 diabetes mellitus as well as (iii) recommendations for self-monitoring glucose for women with gestational diabetes mellitus. Moreover, we added new information about vitamin C supplementation, and insulin administration during pregnancy.
The summary is more or less not existing and must be enhanced, I also recommend to use the abbreviations for the DNA bases.
Response: We have enhanced summary with new information, and abbreviations for the DNA bases.
This review omits many outstanding researchers in oxidative damage that must be cited before this review is published. There are several additional publications required to be cited for elucidation and investigations of oxidative DNA damages.
Response: Thank you for the valid point. We have added researchers in oxidative DNA damage and repair, such as: Sies H., Lindahl T. as well as Cadet J.
Reviewer 2 Report
Line 31 should be “gestational diabetes mellitus”.
Line 35 should be “type 1 or 2 diabetes mellitus”.
Line 112 What is “RAGEs”?
Page 7, Table 1 There should be the abbreviations for GDM and MGH.
Line 302 – summary Do the authors provide the possible cut-off values of 8-OHdG and 8-oxodG for the clinical application for the diagnosis of GDM?
Author Response
Dear reviewer,
Thank you for the review of our paper. We have answered each of the points below.
Line 31 should be “gestational diabetes mellitus”. Line 35 should be “type 1 or 2 diabetes mellitus”.
Response: We have made corrections in all cases you have mentioned
Line 112 What is “RAGEs”?
Response: We have made correction. This abbreviation was explained as Receptors for Advanced Glycation End Products (RAGEs)
Page 7, Table 1 There should be the abbreviations for GDM and MGH.
Response: We had a little bit of confusion with this. There are abbreviations for GDM and MGH. We explained abbreviation under the Table 1.
Line 302 – summary Do the authors provide the possible cut-off values of 8-OHdG and 8-oxodG for the clinical application for the diagnosis of GDM?
Response: It is a very good point that you have made, following your advice we provided possible cut-off values of 8-OHdG for the clinical application for the diagnosis of GDM. We did not add cut-off values of 8-oxodG, because, as explained in the manuscript, 8-OHdG seems to give statistically significant results, test sample is easy to obtain and urinary excretion of 8-OHdG is not associated with human diet.
Reviewer 3 Report
Sandra k et al. performed a narrative review for biomarker for GDM.
This review manuscript encompasses an interesting topic; however, some minor revisions are required for this review.
Comments:
Figure 1 is low resolution. Figure 2; Please provide the name of chemicals, and the explanation for these metabolisms.
Author Response
Dear Reviewer,
Thank you for the review of our paper. We have answered each of the points below.
Sandra k et al. performed a narrative review for biomarker for GDM.
This review manuscript encompasses an interesting topic; however, some minor revisions are required for this review.
Comments:
Figure 1 is low resolution.
Response: Yes, we agree, and we have optimized and improved resolution of Figure 1 to 600 DPI.
Figure 2; Please provide the name of chemicals, and the explanation for these metabolisms.
Response: We have provided the name of chemicals and the explanation for these metabolisms. We do hope that now the Figure 2 will be clearer for readers, and better illustrate scheme of 8-oxo-7,8-dihydro-2-deoxyguanosine, and 8-hydroxydeoxyguanosine generation.
Round 2
Reviewer 1 Report
The authors have revised their review and addressed most of my remarks accordingly. This review is improved and will be an important contribution to the field of DNA damage. The only changes that are required before publication in Molecules are optimized structures in Figure 2 as these are still distorted. Furthermore, the boxes around the structure number should be removed.